# Effect of Mineral Salt Blocks Containing Sodium Bicarbonate or Selenium on Ruminal pH, Rumen Fermentation and Milk Production and Composition in Crossbred Dairy Cows

**DOI:** 10.3390/vetsci8120322

**Published:** 2021-12-11

**Authors:** Hathaichanok Insoongnern, Wuttikorn Srakaew, Tipwadee Prapaiwong, Napongphot Suphrap, Saisamorn Potirahong, Chalong Wachirapakorn

**Affiliations:** 1Department of Animal Science, Faculty of Agriculture and Natural Resources, Rajamangala University of Technology Tawan-ok, Chonburi 20110, Thailand; hathaichanok_in@rmutto.ac.th; 2Department of Animal Science and Fisheries, Faculty of Science and Technology, Nan Campus, Rajamangala University of Technology Lanna, Nan 55000, Thailand; wuttigorn@rmutl.ac.th; 3Department of Animal Production Technology, Faculty of Agro-Industrial Technology, Chantaburi Campus, Rajamangala University of Technology Tawan-ok, Chantaburi 22210, Thailand; tipwadee_pr@rmutto.ac.th; 4Feed Research and Innovation Center (C.P.), Chonburi 20220, Thailand; naphongphot.sup@cp.co.th; 5The Livestock Office of the Region 1, Pathum Thani 12000, Thailand; saisamornpo@gmail.com; 6Department of Animal Science, Faculty of Agriculture, Khon Kaen University, Khon Kaen 40002, Thailand

**Keywords:** mineral salt block, sodium bicarbonate, ruminal pH, selenium, dairy cows

## Abstract

Ruminal pH is an important physiological parameter that regulates microbe activity; optimizing ruminal pH may improve rumen fermentation and milk production. The purpose of this experiment was to determine the effect of sodium bicarbonate (NaHCO_3_) or selenium (Se) in mineral salt block (MSB) supplementation on ruminal pH, rumen fermentation, milk yield and composition in Holstein Friesian crossbred dairy cows. Four crossbred dairy cows with an initial weight of 456 ± 6 kg in mid-lactation were assigned at random using a 4 × 4 Latin square design. The experiments were divided into four periods, each lasting 21 days. Each cow was fed a basal diet supplemented with a different type of mineral salt block: a control with no MSB supplementation, and MSB groups with MSB containing NaHCO_3_ (MSB-Na), MSB containing Se (MSB-Se), and conventional commercial MSB (MSB-Com). MSB-Na contained NaHCO_3_ (500 g/kg) to prevent acidosis, MSB-Se contained organic Se (15 mg/kg) as an antioxidant, and MSB-Com was a positive control mineral salt block. The results show that there was no significant difference in feed intake between treatments, but there was a significant difference in mineral salt intake between treatments (*p* < 0.05). Supplementing mineral blocks had no effect on nutrient intake or apparent digestibility (*p* > 0.05). Ruminal pH was not different between treatments at 0 and 1 h post-feeding, but at 2 and 4 h post-feeding, ruminal pH in cows fed MSB-Na and MSB-Se was significantly higher (*p* < 0.05) than it was in cows fed MSB-Com and the control. Total volatile fatty acid (VFA), acetic, propionic, butyric, and ammonia nitrogen and blood urea nitrogen were not influenced by mineral blocks supplementation. Milk yield, milk composition and energy-corrected milk (ECM) were not affected by supplementing mineral blocks. However, compared with the control, the somatic cell count (SCC) in the milk was reduced (*p* < 0.05) by supplementation with the mineral salt block. Based on the results of the experiments, it was concluded that MSB-Na or MSB-Se supplementation improved ruminal pH while having no effect on feed intake, rumen fermentation, milk yield, or composition, though it did reduce SCC in milk. However, additional research should be conducted to investigate the effect of MSB on rumen ecology and milk production in dairy cows fed a high-concentrate diet.

## 1. Introduction

Minerals, in addition to energy and protein, are important nutrients that play a role in a variety of metabolic activities and are constituents of the body’s composition. Because most minerals are in insufficient concentrations in feeds, animals frequently exhibit signs of insufficiency, resulting in decreased productivity and health. Microelement deficiencies in dairy and beef cattle are frequently observed in temperate areas [1,2]. In tropical areas, Garg et al. [3] and Bhanderi et al. [4] reported that serum samples were below the critical levels of Cu and Zn due to deficiencies of these elements in forages. Microminerals found in forages, such as zinc and copper, are found at levels below the requirements for dairy cows [1]. Even in dairy cows, micromineral deficiency is very common and frequent in tropical conditions. Supplementation by adding minerals to the feed, mineral licks, and/or single/repeated injection of particular microelements, as well as combinations of the these methods, are all options. However, each method has its own set of benefits and drawbacks. Microelement supplementation via access to a mineral salt block is commonly used in dairy farms. A mineral salt block is a deposit of macro and micro minerals that animals utilize to supplement their diets and ensure that they obtain enough minerals.

Ruminal pH is the most important rumen ecological factor affecting the final product produced by microbial fermentation and can be used by ruminants for productivity and health. The ideal ruminal pH is between 6.3 and 7.0, where the most efficient fermentation can be achieved. Cellulolytic bacteria in the rumen cannot grow at a pH lower than 6.0 and a moderate increase in rumen pH will promote their activity [5]. Mild acidosis occurs when the ruminal pH falls below 6.0, while subacute acidosis occurs when the pH drops below 5.5. Subacute ruminal acidosis (SARA) is described as periods of moderately low ruminal pH that are between acute and chronic in duration [6]. SARA is predominantly caused by feeding high-concentrate-based diets; the most constant and early clinical symptom of SARA is reduced feed intake, presumably due to excess organic acids disrupting rumen function [7]. To overcome this, sodium bicarbonate (NaHCO_3_) is widely used as an effective buffer in the rumen to maintain ruminal pH at an optimum level [8,9]. Sodium bicarbonate can be used by adding it to the ration, by free choice or by incorporation into a mineral salt block. Ichijo et al. [10] demonstrated that licking a block-type agent that contained NaHCO_3_ reduced the ruminal pH of SARA-affected steers more effectively than the oral administration of the same amount of NaHCO_3_ powder.

Subclinical mastitis in lactating dairy cows often occurs on poorly managed dairy farms. In addition, subclinical mastitis can cause a decrease in milk yield (10–20%) and quality, as well as an increase in the somatic cell count (SCC) [11,12]. In order to reduce such incidence, there are many approaches to prevent mastitis by using antibiotics, phytonutrients, plant extracts, etc. Microelements, such as zinc (Zn), on the other hand, play a role in optimizing the cellular immune response [13] and the development of keratin, which entraps these bacteria [14], so including adequate Zn in the dairy cow diet may help reduce SCC [15,16]. In addition, Zn, copper (Cu), and selenium (Se) supplementation has been associated with the higher antioxidant capacity of superoxide dismutase (CuZn-SOD), glutathione peroxidase (GSH-Px), and serum ceruloplasmin (CP), respectively, resulting in SCC [17]. Se is an essential component of GSH-Px, which is responsible for the conversion of hydrogen peroxide (H_2_O_2_) and free oxygen (O_2_) to water (H_2_O) in the antioxidant system [18]. GSH-Px activity in milk catalyzes the reduction of various peroxides, protecting the cell from oxidative damage [19]. The administration of Se to periparturient cows reduces the occurrence and severity of mastitis [20].

Mineral salt blocks have a variety of ingredients, depending on the objectives of animal use, such as improving fertility, reducing SCC, and optimizing ruminal pH. There are few studies on feeding NaHCO_3_ or Se mineral salt blocks to dairy cows under tropical conditions. Our knowledge of mineral salt blocks is based primarily on very limited data. Therefore, the purpose of this study was to determine the effect of mineral salt blocks containing NaHCO_3_ or Se as a dietary supplement on ruminal pH, dry matter intake, digestibility, milk production, and SCC in crossbred dairy cows.

## 2. Materials and Methods

### 2.1. Animal Welfare Statement

The experiment was reviewed and approved by the Animal Ethics Committee of Khon Kaen University (permission No. ACUC-KKU 70/2559), based on the Ethics of Animal Experimentation of National Research Council of Thailand.

### 2.2. Animals, Diets, Experimental Design and Treatments

This study was carried out at the Dairy Unit, Faculty of Agriculture, Khon Kaen University, Thailand. Animals were chosen based on their body weight (BW), milk yield, and day in milk (DIM), and they were handled with care to avoid unintentional errors. Four lactating Holstein Friesian crossbred cows with initial body weight 456 ± 6 kg, average milk yield 11 ± 2 kg and in mid-lactation (102 ± 34 DIM) were allotted into four dietary treatments according to a 4 × 4 Latin square design. Animals were housed individually in a concrete floor pen (4 × 4 m^2^) with their own feed bunk and clean water tank. Animals received the same diets and were supplemented with different mineral salt blocks: the control (non-supplement) and the mineral salt block groups. Mineral salt blocks were classified as follows: MSB-Na containing NaHCO_3_ (500 g/kg), which was used to avoid rumen acidosis, MSB-Se had organic Se (15 mg/kg), which was used as an antioxidant, and MSB-Com was used as a positive control mineral salt block, which is a commonly accessible commercial salt block. The mineral composition of each mineral salt block is shown in Table 1.

### 2.3. Data Record, Sampling Procedures and Analysis Methods

This experiment was carried out for four periods consisting of 21 days in each period. The first 14 days were for animal adaptation, and the last 7 days were for sample collection. The feeding times were 7:00 a.m. and 3:00 p.m. Rice straw as roughage was offered ad libitum while concentrates were offered depending on milk yield at a concentrate-to-milk ratio of 1:1.5. Nutrient intake of animals was calculated according to their body weight and milk production [18]. Intake of rice straw and concentrate was recorded for 21 days, and the last 7 days were devoted to collection of samples for analysis. Dry matter intake was calculated by using the dry matter of rice straw and concentrate. The mineral salt block was available at all times. Intake of minerals was subtracted by weight at the beginning and end of each period. The chemical compositions of the rice straw and concentrate are shown in Table 2.

The samples of feeds including concentrate and rice straw and samples of feces were taken during the last 7 days of each period. The first half of the samples was analyzed daily for DM content, while the other half of the samples was pooled and kept at −20 °C for chemical composition analysis. Feces samples were collected using rectal palpation to avoid contamination with soil and were kept frozen. The frozen feed, refusal, and fecal samples were thawed and oven-dried at 60 °C for 72 h. Then, all samples were ground through a 1 mm screen and analyzed for their chemical composition including crude protein (CP), ether extract (EE), and ash using methods used by the AOAC [21]. Contents of lignocellulose (acid detergent fiber, ADF) and cell walls (neutral detergent fiber, NDF) were analyzed using the technique of Van Soest et al. [22]. Acid insoluble ash (AIA) was analyzed and used as an indicator using 2N HCl according to the method of Van Keulen and Young [23] and calculated for the apparent digestibility [24] as follows: DM digestibility, % = 100 − [100 × (%AIA in feed) × (%AIA in feces)]; Nutrient digestibility, % = 100 − [(100 × %AIA in feed ÷ %AIA in feces) × (%nutrient in feces ÷ %nutrient in feed)].

On the last day of each period (day 21), rumen fluid and blood samples of each animal were collected at 0 h post-feeding (before morning feeding), 1 h post-feeding, 2 h post-feeding, and 4 h post-feeding on the last day of each collection period. Approximately, rumen fluid (200 mL) from each animal was taken via a stomach tube attached to a vacuum pump. Ruminal pH was measured immediately using a pH meter (Hanna instrument HI 8424 microcomputer, Singapore) and subsequently filtered through four layers of cotton cheesecloth. Forty-five milliliters of rumen fluid was mixed with 5 mL of 1 M H_2_SO_4_ and used for ammonia nitrogen (NH_3_-N) and volatile fatty acids (VFA) analysis. The NH_3_-N concentration was analyzed by using the Kjeltech Auto 1030 Analyzer according to the AOAC [21]. The VFA concentration was analyzed using high pressure liquid chromatography (HPLC) (instruments by water and Novapak model 600E; water mode 1484UV detector; column Novapak C18; column size 3.9 × 300 mm; mobile phase 10 mM H_2_PO_4_ (pH 2.5)) according to the technique of Cai [25]. The blood sample was taken from the jugular vein—10 mL from each animal transferred into tubes containing 12 mg of ethylene diamine tetra-acetic acid (EDTA)—and immediately stored on ice and brought back to the laboratory for analyzing blood urea nitrogen (BUN) and blood glucose using an autoanalyzer.

The milk yield of each cow was recorded daily for 21 days and milk samples were collected twice daily on the last 7 days of each period; the milk sample ratio of the morning and afternoon milk samples was 60:40. Milk samples were divided in two parts. The first part was composed daily and preserved with potassium dichromate (K_2_Cr_2_O_7_) and then stored at 4 °C for analyzing milk components using Milko-Scan 104 (Foss Electric, Denmark). The second part was analyzed for SCC using a Fossomatic 500 Basic (Foss Electric, Integrated Milk Testing^TM^).

### 2.4. Statistical Analysis

Data were statistically analyzed according to a 4 × 4 Latin square design using the procedure of general linear models in SAS software [26]. The means were further tested using the Duncan’s new multiple range test. The difference was declared at *p* < 0.05. Trends were declared at *p* > 0.05 to *p* ≤ 0.10.

## 3. Results

### 3.1. Feed Intake and Nutrient Digestibility

Table 3 shows the effects of different mineral salt blocks on feed intake and nutrient digestibility. The mineral salt block had no effect on feed intake and apparent nutrient digestibility (*p* > 0.05) compared to the control. The mineral salt block intake of MSB-Se was higher than that of MSB-Com (*p* < 0.05), but there was no difference (*p* > 0.05) with MSB-Na.

### 3.2. Milk Yield and Composition

Table 4 shows the effect of mineral block supplementation on milk yield and quality in lactating dairy cows. The results showed that milk yield and 4% FCM, as well as milk composition (fat, protein, lactose, non-fat solids, and total solids) did not differ between treatments (*p* > 0.05). When compared to the control, mineral salt block supplementation reduced SCC (*p* = 0.07) and the somatic cell score (SCS) (*p* < 0.05). SCC and SCS, on the other hand, did not differ between mineral salt block supplementation groups (*p* > 0.05).

### 3.3. Ruminal Fermentation Parameters and Blood Urea Nitrogen

Table 5 presents the characteristics of rumen fermentation, such as pH, NH_3_-N, volatile fatty acids (VFA), and blood metabolites such as BUN and glucose. The levels of NH_3_-N, BUN, and glucose did not differ (*p* > 0.05) between treatments. Total VFA levels and individual VFA profiles did not differ (*p* > 0.05) between treatments. Ruminal pH at 0 and 1 h post-feeding did not differ (*p* > 0.05) between treatments. However, ruminal pH at 2 and 4 h post-feeding was higher (*p* < 0.05) in cows supplemented with MSB-Na and MSB-Se than it was in cows fed MSB-Com and the control (Figure 1).

## 4. Discussion

### 4.1. Feed Intake and Nutrient Digestibility

In this study, the mineral salt blocks had no effect on nutrient intake. However, cows were given concentrate and rice straw with the composition, as previously described, and they acquired CP as suggested by Wachirapakorn et al. [30], who reported that lactating crossbred cows producing 11–13 kg of milk required 12–14 percent crude protein in their diet. In comparison to the control, the mineral salt block group had similar apparent digestibility. These findings are in line with a report by Raucha et al. [9], who found no difference in apparent digestibility in cows supplemented with NaHCO_3_ or calcium magnesium carbonate. Wittayakun et al. [31] reported that effective fiber (long form rice straw) or NaHCO3 supplementation had no effect on feed intake or nutrient digestibility in dairy cows fed a 70:30 pineapple peel–concentrate pellet mixed diet. In this study, rice straw was used as the sole roughage and acted as an effective fiber, which may be a key factor influencing intake and digestibility in cows. Zhao [32] revealed that long rice straw has no effect on intake or digestibility but increases rumination and chewing activity when compared to short or medium rice straw.

Mineral salt block intake, on the other hand, was higher in MSB-Se than in MSB-Com, but not in MSB-Na. The palatability or components of mineral salt blocks may lead to a wide range of intakes. According to Chládek and Zapletal [33], the daily intake of mineral blocks in grazing beef cows varied depending on the components of the mineral blocks; however, the overall intake of mineral blocks was 24.6 and 28.1 g/d during winter and the pasture grazing period, respectively. Furthermore, the difference in the amount of mineral salt blocks consumed could be attributed to the Na concentration in the blocks. In comparison to MSB-Na and MSB-Com, the amount of Na in MSB-Se was the lowest, so animals had to consume more to meet their Na requirements. Thiangtum et al. [34] recommended that dairy cows in tropical conditions require Na 1.2 g/kg DM.

### 4.2. Milk Yield and Composition

Mineral salt block supplementation had no effect on milk yield and milk composition. Mineral salt blocks, on the other hand, dramatically reduced SCC and SCS in milk when compared to the control. Because of the important microelements in the blocks, SCC and SCS are reduced. Zn was one of the microelements in the block that has been shown to minimize SCC in milk [2,35]. Although, plasma Zn concentrations were not assessed in this study, Pechová et al. [35] found no significant differences in plasma Zn concentrations between the experimental group supplemented with Zn in the chelate form at a dose of 2.2 g per animal per day and the control group. Davidov et al. [2] assessed blood Zn concentrations and SCC in dairy cows at different stages of lactation, and indicated that with increasing levels of Zn in the blood, a decline in SCC and intramammary infections was observed. Therefore, it could be concluded that Zn has an effect on milk SCC in dairy cows. Recently, Bakhshizadeh et al. [36] showed that Zn supplementation in the form of zinc nano (ZnN) and zinc glycine (ZnGly) decreased SCC compared with zinc oxide (ZnO) and unsupplemented treatments. These results are in accordance with those obtained by Dibley [37] who reported that Zn plays an integral role in immune function by activating T-lymphocyte responsiveness, thus impacting the effectiveness of somatic cells within the mammary gland. This is thought to be due in part to Zn’s role in the formation of the keratin lining of the teat and the protection it provides against bacterial infection, as well as Zn’s extensive influence on immunity function and inflammation [38].

Mineral salt blocks MSB-Se and MSB-Com in this current study contained Zn, Cu, and Se, while MSB-Na contained Zn and Cu. The contribution of lower SCC in milk is likely due to two reasons: first, Zn is involved in the formation of keratin, which traps bacteria as they enter the udder [14], and in optimizing cellular immunological response [13,37]; second, Zn, Cu, and Se may combine to generate CuZn-SOD, GSH-Px, and CP, which may reduce cell damage owing to bacterial attack in the udder [17]. Machado et al. [39] found that injecting a trace mineral mixture (Zn, Mn, Se and Cu) during pre-partum and postpartum periods enhanced SOD activity in cows but did not affect leukocyte function. Warken et al. [40] used a mineral complex supplement (Mg, P, K, Se and Cu) in early lactating dairy cows that reduced SCC; they observed that SOD activity and cytokine level increased in supplemented cows compared to the control. Furthermore, Colakoglu et al. [41] discovered a negative correlation between GSH-Px activity in milk and SCC; thus, Se in MSB-Se is thought to be due to the actions of certain antioxidant Se-dependent enzymes that reduce SCC in milk [42].

### 4.3. Ruminal Fermentation Parameters

The ruminal NH_3_-N concentration was similar in all dietary treatments since the animals had received the same CP content and apparent digestibility. Likewise, the concentrations of TVFA and individual VFA were not influenced by mineral salt block supplementation. This study found that MSB-Na had no effect on rumen fermentation, which is consistent with Wittayakun et al. [31], who supplement 1.2% NaHCO3 in the diet of dairy cows.

The average ruminal pH found in this current study was in the optimum pH range (6.3 to 7.2) for rumen microorganism growth, as reported by Yang and Beauchemin [43]. Wanapat et al. [44] found that as the ratio of concentrate to roughage in dairy steer diets increased from 20:80 to 80:20, ruminal pH decreased linearly. Ruminal pH was found to be 6.2 in steers fed a diet with a concentrate-to-roughage ratio of 40:60, which was lower than that of the current study. Ruminal pH in this study may have had no effect on fibrolytic bacteria, resulting in a lower response in rumen fermentation.

At 4 h after feeding, animals receiving mineral salt block MSB-Na had a higher value of ruminal pH than the control animals. This result might be due to the fact that MSB-Na had a higher proportion of NaHCO_3_; as a result, mineral salt block MSB-Na supplementation increased ruminal pH, as expected. When compared to the values after NaHCO_3_ oral administration, Ichijo et al. [10] found that the mean ruminal pH after licking a salt block containing NaHCO_3_ was significantly higher and that diurnal fluctuations in ruminal pH tended to be more stable. However, ruminal pH in cows fed the mineral salt block MSB-Se was similar to that of cows fed with MSB-Na, although the cause for this was unclear. Higher intake of MSB-Se, on the other hand, may result in more licking and stimulation of salivation, thereby improving ruminal pH. According to Abe [45], cows given a mineral salt block similar to MSB-Se increased salivation by 150%.

The amount of concentrate intake by the cows in this study could not induce SARA. Krause et al. [46] induced SARA by adding more soluble degraded starch; ruminal pH was found to be 5.69, and cows with access to the buffer blocks had a lesser fall in mean ruminal pH during the SARA challenge and recovered more quickly. Cows with SARA were provided access to NaHCO_3_ water, but their ruminal pH remained at the SARA level [47]. The approach of introducing buffer salt blocks to cows is thought to be effective. According to Mao et al. [48], NaHCO_3_ supplementation increased the final pH levels and TVFA concentrations. Furthermore, pyrosequencing of the 16S rRNA gene revealed that the addition of NaHCO_3_ increased the bacterial diversity index when compared to the control. The addition of NaHCO3 reduced the relative abundance of *Streptococcus* and *Butyrivibrio* while increasing the proportions of *Ruminococcus*, *Succinivibrio*, and *Prevotella*. In addition, Ramos et al. [49] demonstrated that introducing combined buffer agents (calcium oxide, processed coral, magnesium oxide and NaHCO_3_) stabilized the ruminal pH and improved rumen fermentation by changing bacteria communities in dairy cows fed a high-concentrate diet, which may contribute to ruminal acidosis prevention.

## 5. Conclusions

According to the current findings, ruminal pH can be stabilized in the optimal range for rumen microbial activity and milk quality can be improved by decreasing somatic cells in milk by providing mineral salt blocks containing NaHCO_3_ or Se. However, the current findings will need to be confirmed with a high-producing lactating dairy cow fed a high-concentrate diet, as well as with an investigation into whether a mineral salt block could help maintain ruminal pH or increase milk output.

## Figures and Tables

**Figure 1 vetsci-08-00322-f001:**
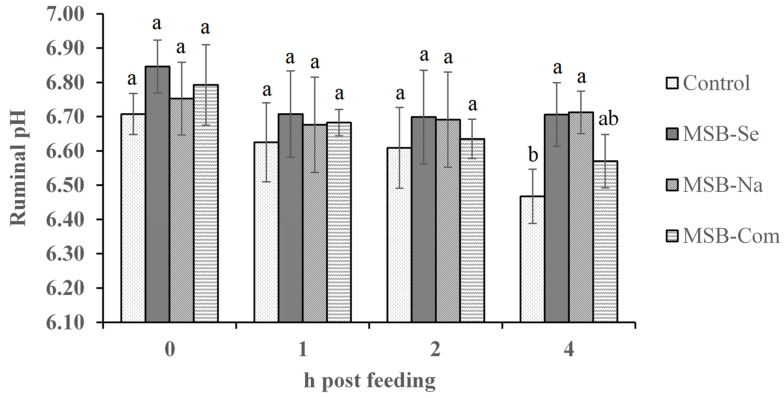
Ruminal pH in cows supplemented with different mineral salt blocks at different hours post-feeding. Control = no mineral salt block (MSB) supplementation, MSB-Se = MSB containing Se supplementation, MSB-Na = MSB containing NaHCO_3_ supplementation, MSB-Com = conventional commercial MSB supplementation. ^a,b^ Means (±se) with different superscript letters were significantly different (*p* < 0.05).

**Table 1 vetsci-08-00322-t001:** Chemical composition of mineral salt blocks.

Composition	Unit	Mineral Salt Block (per 1 kg)		
		MSB-Se	MSB-Na	MSB-Com
Copper, Cu	mg	150	150	450
Cobalt, Co	mg	25	25	60
Ferrous, Fe	mg	-	-	2100
Iodine, I	mg	50	-	150
Manganese, Mn	mg	500	630	420
Selenium, Se	mg	15	-	10
Zinc, Zn	mg	500	620	280
Phosphorus, P	g	-	-	100
Sodium, Na	g	382	-	210
Calcium, Ca	g	-	-	80
Magnesium, Mg	g	-	10	2.5
Salt	g	-	390	-
Molasses	g		18	-
Sodium bicarbonate, NaHCO_3_	g	-	500	-

**Table 2 vetsci-08-00322-t002:** Ingredients used in the concentrate and chemical composition of concentrate and rice straw.

Item	Concentrate	Rice Straw
Concentrate ingredients		
Cassava chip	47.0	
Corn meal	7.0	
Soybean meal	20.0	
Fined rice bran	5.0	
Palm kernel meal	9.5	
Bean pods meal	4.0	
Sugar	3.0	
Urea	2.5	
Salt	0.5	
Dicalcium phosphate	1.0	
Premix ^†^	0.5	
Total	100.0	
Chemical composition, %DM		
DM	90.08	91.97
Crude protein	17.62	2.84
Ether extract	3.48	1.77
NDF	16.11	90.52
ADF	10.24	49.63
Ash	4.42	14.05
ME, Mcal/kg DM	2.79	1.54

DM = dry matter, NDF = neutral detergent fiber, ADF = acid detergent fiber, ME = metabolizable energy. ^†^ Premix consisted of vit. A 10,000,000 IU/kg, vit. E 70,000 IU/kg, vit. D 1,600,000 IU/kg; Fe 50,000 mg/kg, Zn 40,000 mg/kg, Mn 40,000 mg/kg, Co 100 mg/kg, Cu 10 mg/kg, Se 100 mg/kg, I 500 mg/kg.

**Table 3 vetsci-08-00322-t003:** Effects of mineral salt block supplementation on intake and apparent digestibility.

Item		Mineral Salt Block			SEM	*p*-Value	
	Control	MSB-Se	MSB-Na	MSB-Com		C vs. MSB	T
Average BW, kg	464.94	467.46	465.61	469.21	3.762	0.72	0.87
Intake, kg/d							
Concentrate	9.41	9.03	9.30	9.11	0.697	0.76	0.98
Rice straw	5.25	5.72	5.46	5.27	0.150	0.21	0.19
Total	14.66	14.75	14.76	14.38	0.780	0.98	0.98
%BW	3.15	3.06	3.18	3.15	0.166	0.95	0.97
g/kgBW^0.75^	146.23	146.72	147.39	142.61	7.712	0.96	0.97
R:C ratio	35.96	39.02	37.16	36.85	1.471	0.45	0.55
Mineral block lick, g/d	0.00 ^a^	21.43 ^b^	14.29 ^bc^	11.90 ^c^	1.943	**	**
Nutrient intake, kg/d							
OM	13.50	13.56	13.59	13.24	0.737	0.97	0.98
CP	1.81	1.75	1.79	1.76	0.126	0.79	0.98
EE	0.42	0.41	0.42	0.41	0.026	0.85	0.94
NDF	6.27	6.64	6.44	6.26	0.214	0.51	0.59
ADF	3.58	3.76	3.66	3.55	0.122	0.59	0.65
Nutrient digestibility, %							
DM	60.58	58.56	60.42	59.57	1.336	0.39	0.63
OM	64.00	61.34	63.25	62.51	1.411	0.35	0.61
CP	71.78	67.50	70.00	69.22	1.311	0.09	0.32
EE	86.57	83.89	84.28	83.48	1.150	0.11	0.24
NDF	47.22	49.23	48.56	49.00	0.897	0.15	0.45
ADF	43.57	43.39	41.40	43.82	2.965	0.84	0.93
Energy intake							
Mcal ME/d	32.93	31.64	32.61	31.69	2.388	0.74	0.97
Microbial crude protein							
kg/d	1.25	1.08	1.12	1.09	0.081	0.75	0.97

OM = organic matter, CP = crude protein, EE = ether extract, NDF = neutral detergent fiber, ADF = acid detergent fiber, ME = metabolizable energy, microbial crude protein (MCP) = 0.133 × DOMI [27]. C vs. MSB = control vs. mineral salt block group, T = treatments, SEM = standard error of the mean. ^a,b,c^ Means in the same row with different superscript letters were significantly different (*p* < 0.05). ** *p* < 0.01.

**Table 4 vetsci-08-00322-t004:** Effects of mineral salt block supplementation on milk yield and milk production.

Item		Mineral Salt Block			SEM	*p*-Value	
	Control	MSB-Se	MSB-Na	MSB-Com		C vs. MSB	T
Milk production							
Milk yield, kg/d	12.95	13.06	12.46	12.21	0.292	0.31	0.23
4%FCM	12.72	13.49	12.47	12.56	0.619	0.87	0.66
ECM ^†^, kg	12.97	13.48	12.70	12.64	0.537	0.96	0.69
Milk composition							
Fat, %	3.84	4.19	4.04	4.15	0.236	0.34	0.74
Protein, %	3.48	3.40	3.56	3.40	0.104	0.81	0.66
Lactose, %	5.14	5.09	5.12	5.11	0.024	0.28	0.60
Solid-not-fat, %	9.32	9.19	9.37	9.21	0.118	0.68	0.66
Total solids, %	13.16	13.37	13.41	13.36	0.278	0.51	0.91
Fat/protein ratio	1.11	1.22	1.14	1.23	0.073	0.32	0.58
Milk efficiency, kg/kg DM	0.87	0.89	0.84	0.86	0.051	0.88	0.87
4%FCM/kg DM	0.86	0.93	0.84	0.89	0.071	0.77	0.83
NUE ^‡^	24.05	24.86	23.76	23.74	2.195	0.98	0.98
SCC, ×10^3^ cell/ml	319.50	132.25	114.75	139.00	48.82	*	0.07
SCS ^§^	4.54 ^a^	3.24 ^b^	3.09 ^b^	3.18 ^b^	0.265	**	0.02

4% FCM = 4% fat-corrected milk. SNF = solids-not-fat, SCC = somatic cell count, ECM = energy-corrected milk. ^†^ ECM = milk × (0.38 × % fat + 0.24 × % protein + 0.17 × % lactose)/3.17. ^‡^ Nitrogen utilization efficiency (NUE) = ((milk protein yield, kg/day) ÷ 6.38)/((CP intake, kg/day) ÷ 6.25) [28]. ^§^ Somatic cell score (SCS) = log_2_ (SCC/100,000) + 3 [29]. C vs. MSB = control vs. mineral salt block group, T = treatments, SEM = standard error of the mean. ^a,b^ Means in the same row with different superscript letters were significantly different (*p* < 0.05). * *p* < 0.05, ** *p* < 0.01.

**Table 5 vetsci-08-00322-t005:** Effects of mineral salt block supplementation on rumen fermentation and blood metabolites.

Item		Mineral Salt Block			SEM	*p*-Value	
	Control	MSB-Se	MSB-Na	MSB-Com		C vs. MSB	T
Rumen end-products							
pH	6.60	6.74	6.71	6.67	0.048	0.11	0.30
NH_3_-N, mg/dL	17.20	19.50	18.16	18.81	0.689	0.09	0.21
TVFA, mM	97.44	99.34	97.58	98.40	0.831	0.34	0.42
C2, %	67.80	67.66	67.79	67.51	0.030	0.69	0.87
C3, %	21.13	21.18	21.06	21.31	0.274	0.87	0.93
C4, %	11.08	11.16	11.15	11.18	0.090	0.43	0.86
C2/C3	3.39	3.40	3.38	3.31	0.062	0.70	0.73
Blood metabolites							
BUN, mg/dL	19.56	18.75	19.69	19.81	1.045	0.91	0.89
Glucose, mg/dL	58.25	55.63	55.88	57.69	1.371	0.29	0.49

C vs. MSB = control vs. mineral salt block group, T = treatments, SEM = standard error of the mean.

## Data Availability

The data that support the findings of this study are available from the corresponding author, C.W., upon reasonable request.

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
