# Peer review of "Effect of Mineral Salt Blocks Containing Sodium Bicarbonate or Selenium on Ruminal pH, Rumen Fermentation and Milk Production and Composition in Crossbred Dairy Cows"

_vetsci, 2021, doi:10.3390/vetsci8120322_

Round 1
Reviewer 1 Report
First of all, congratulations on this work.
However, I am finding a critical point that would need to be further addressed. The animal sample is so reduced, so that the effect of age and number of parturitions per animal are being disregarded.
In the conclusions section, I suggest to improve its content.
Reviewer 2 Report
The manuscript is based on a well planned and described experiment, results are in general clear and the introduction and discussion are well focused. However, I would like to re-evaluate this manuscript once some methodological clarifications will be done.
The description of the mineral blocks is not as clear as it seems… And the blocks should be perfectly understood in order to evaluate the discussion and conclusions.
- What is the control block made of?
- The K block just contains microminerals, in the order of mg. What is the rest of the block made of? More important, why is it repeatedly called “the block with Se” when it has just 15 mg? (the block C contains 10 mg…) And why does it include 385 mg of Na, which, first, it is a micromineral and, second, the amount is ridiculous for such a limiting mineral for ruminants?
- The C block (macro+microminerals) is also unclear. I understand that dot is used for decimals and comma for thousands. Thus, I guess the unit for Mg should be mg, otherwise the block would contain 2.5 kg of Mg per kg! Once corrected, it makes around 400 g, so, again, what is the rest of the block made of?
Then, I would like to see some further explanation about how were the feed intake and digestibility calculated (missing in the text).
In the result tables, what means C vs M?
Finally, I would like to see a different title, since the current one does not adequately reflect the main results.
Without clarification of the points above, it is not possible for me to make an adequate evaluation of the paper, so I will be happy to do it once all this will be clarified.
Author Response
Response to Reviewer Comments
Reviewer# 2
Point 1: The manuscript is based on a well-planned and described experiment, results are in general clear and the introduction and discussion are well focused. However, I would like to re-evaluate this manuscript once some methodological clarifications will be done.
Response 1:
We would like to thank you for taking the time to review our manuscript. We did our best to respond to your comments.
Point 2: The description of the mineral blocks is not as clear as it seems… And the blocks should be perfectly understood in order to evaluate the discussion and conclusions.
Response 2:
Thank you very much for your insightful comments. We hypothesized that mineral blocks containing sodium bicarbonate or selenium could improve rumen fermentation and milk production by stabilizing ruminal pH in our study.
Point 3: What is the control block made of?
Response 3:
We had four treatments, with no supplemented MSB serving as the control and commercial MSB serving as the positive control. The other two MSBs contained sodium bicarbonate (NaHCO3) and selenium, with both aiming to improve ruminal pH. The control block is made of commercial mineral block, which is widely used by dairy farmers in our area.
Point 4: The K block just contains microminerals, in the order of mg. What is the rest of the block made of? More important, why is it repeatedly called “the block with Se” when it has just 15 mg? (the block C contains 10 mg…) And why does it include 385 mg of Na, which, first, it is a micromineral and, second, the amount is ridiculous for such a limiting mineral for ruminants?
Response 4:
We sincerely apologize for our error. We have already double-checked the amount of minerals in the block and corrected and rearranged the order. Please see Table 1 for more information. After accumulating a certain amount of minerals, the remainder is carriers, which is difficult to determine because it is dependent on the manufacturer.
Mineral slat blocks of type A, K, and C have been replaced with MSB-Na, MSB-Se, and MSB-Com. MSB-Na is a mineral salt block that contains sodium bicarbonate, MSB-Se is a mineral salt block that contains selenium, and MSB-Com is a mineral salt block that is commonly used by dairy farmers and is available in our area.
Point 5: The C block (macro+microminerals) is also unclear. I understand that dot is used for decimals and comma for thousands. Thus, I guess the unit for Mg should be mg, otherwise the block would contain 2.5 kg of Mg per kg! Once corrected, it makes around 400 g, so, again, what is the rest of the block made of?
Response 5:
Once again, it is our fault. The amount and unit of minerals in the block were corrected. Please refer to Table 1. Thank you for your concern, as we already stated in point 4.
Point 6: Then, I would like to see some further explanation about how were the feed intake and digestibility calculated (missing in the text).
Response 6:
This section has been added to Materials and Methods to make it more clear. Please see the text for more information.
Point 7: In the result tables, what means C vs M?
Response 7:
We have changed ‘C vs M’ to ‘C vs MSB’ in Table 2, 3, 4, and 5. C vs MSB refers to a comparison of the means of the control group and the MSB groups. T denotes a comparison of four treatments.
Point 8: Finally, I would like to see a different title, since the current one does not adequately reflect the main results.
Response 8:
We have changed title as suggested by the reviewer, please see in the text.
Point 9: Without clarification of the points above, it is not possible for me to make an adequate evaluation of the paper, so I will be happy to do it once all this will be clarified.
Response 9:
Thank you very much for your time; we hope that the manuscript will be satisfactory to you after revision based on the reviewer's comments.

Reviewer 3 Report
Comments and Suggestions for Authors
The manuscript presents the effect of the mineral salt blocks containing sodium bicarbonate or selenium on ruminal pH and volatile fatty acid, milk production and composition, and blood metabolites. The study deals with an interesting subject and it could be better if the authors evaluated the milk fatty acids that vary according to the ruminal fermentation and biohydrogenation. However, the manuscript should be improved.
Title:
I suggest changing the title. The title presents just MSB with sodium bicarbonate without citing MSB with selenium. Also, MSB with sodium bicarbonate alters ruminal pH and Somatic Cell counts without an effect on Milk production and composition.
Abstract:
The abstract should contain a brief introduction to explain the interest of MSB use and the inclusion of sodium bicarbonate or selenium.
Material and methods
This section could be improved.
Line 115: It is simpler and clearer to subdivide the subsection into several others for example (diet analysis; ruminal fermentation analysis; blood metabolites….)
Results
The first paragraph should be deleted. The diet composition of the experiment is not a result. The Tables 1 and 2 should be reported in Material and Methods.
Table 4. It could be better to add superscript to highlight the effect of MSB vs control.
Discussion
The discussion needs a deep improvement. The discussion is a little bit superficial and does not link the results obtained, and there is a repetition of the results.
It was preferable and more logical to discuss feed intake, ruminal parameters and production and other parameters.
Conclusion
It is mandatory to change the conclusion. Do not use block types, it is better to use blocks containing sodium bicarbonate or selenium. It does not shed light on the interest of the results obtained and the interest in using blocks with additives, and also it does not link them to the initial hypothesis reported in the introduction.
Author Response
Response to Reviewer Comments
Reviewer# 3
Point 1: The manuscript presents the effect of the mineral salt blocks containing sodium bicarbonate or selenium on ruminal pH and volatile fatty acid, milk production and composition, and blood metabolites. The study deals with an interesting subject and it could be better if the authors evaluated the milk fatty acids that vary according to the ruminal fermentation and biohydrogenation. However, the manuscript should be improved.
Response 1:
We greatly appreciated your positive feedback. We're sorry, but milk fatty acids have not been studied. We attempted to change, add, and rewrite in response to the reviewer's comments, and we hope it improved as you intended.
Point 2: Title:
I suggest changing the title. The title presents just MSB with sodium bicarbonate without citing MSB with selenium. Also, MSB with sodium bicarbonate alters ruminal pH and Somatic Cell counts without an effect on Milk production and composition.
Response 2:
As you suggested, we changed the title.
The title has been changed to "Effect of mineral salt blocks containing sodium bicarbonate or selenium on ruminal pH and fermentation, milk production, and composition in crossbred dairy cows."
Point 3: Abstract:
The abstract should contain a brief introduction to explain the interest of MSB use and the inclusion of sodium bicarbonate or selenium.
Response 3:
We have already included a brief introduction in the abstract. Please see the text for more information.
Point 4: Materials and Methods:
This section could be improved.
Response 4:
We rewrote the Materials and Methods section to make it more understandable.
Point 5: Line 115: It is simpler and clearer to subdivide the subsection into several others for example (diet analysis; ruminal fermentation analysis; blood metabolites….)
Response 5:
We rewrote and divided or grouping such as intake measurement, diet analysis and digestibility calculation, rumen fluid and blood samplings, and analysis by paragraph. Please see the text for more information.
Point 6: Results:
The first paragraph should be deleted. The diet composition of the experiment is not a result. The Tables 1 and 2 should be reported in Material and Methods.
Response 6:
As suggested, we deleted. As per your suggestion, Tables 1 and 2 were moved to Materials and Methods.
Point 7: Table 4. It could be better to add superscript to highlight the effect of MSB vs control.
Response 7:
We looked at the differences between the control and the MSB. There was no difference if the P-value was greater than 0.05, and there was a significant difference if the P-value was less than 0.05.
Point 8: Discussion:
The discussion needs a deep improvement. The discussion is a little bit superficial and does not link the results obtained, and there is a repetition of the results.
Response 8:
We rewrote to improve the discussion by including more references, and then rewrote again to reduce repetition. Thank you very much for your constructive criticism.
Point 9: It was preferable and more logical to discuss feed intake, ruminal parameters and production and other parameters
Response 9:
As suggested by the reviewer, we have added more references to the discussion. Please see the text for more information.
Point 10: Conclusion:
It is mandatory to change the conclusion. Do not use block types, it is better to use blocks containing sodium bicarbonate or selenium. It does not shed light on the interest of the results obtained and the interest in using blocks with additives, and also it does not link them to the initial hypothesis reported in the introduction.
Response 10:
As a result of your feedback, we revised the conclusion. Please see the text for more information.
Finally, we'd like to thank you so much for your insightful comments, which helped to improve the readability of our manuscript.

Round 2
Reviewer 1 Report
Congratulations! The papers has been substantially improved, especially the discussion.
Author Response
Response to Reviewer Comments
Reviewer# 1
Point 1:
Congratulations! The papers has been substantially improved, especially the discussion.
Response 1:
We would like to express our heartfelt gratitude for your assistance.
Please see the attachment.

Reviewer 2 Report
I´m satisfied with the changes implemented in the manuscript.
Author Response
Response to Reviewer Comments
Reviewer# 2
Point 1:
I´m satisfied with the changes implemented in the manuscript.
Response 1:
We appreciate your assistance and would like to thank you for reviewing our manuscript.
Please see the attachment.

Reviewer 3 Report
Comments and Suggestions for Authors
The manuscript presents the effect of the mineral salt blocks containing sodium bicarbonate or selenium on ruminal pH and volatile fatty acid, milk production, and composition, and blood metabolites. The study deals with an interesting subject. I appreciate the modification done by the authors; however, I think that the authors focus more on Sodium Bicarbonate and neglected Selenium and, the introduction and the conclusion should be improved.
Abstract:
Line 18: The added introduction should present the importance to study milk production.
Line 26: I think it is more convenient to change general by conventional.
Line 40: authors should add a conclusion which presents the application of the results and recommendation.
Keywords: Selenium was not mentioned (authors focused on Bicarbonate de sodium more than Selenium)
Introduction:
The introduction should be improved. The authors did not present the interest in selenium and did not justify why they chose to add selenium.
Material and methods
Line 100: in which country.
Line 166: “was analyzed” was reported twice in the sentence.
Results
Line 180: delete the second “milk”
Line 185: the authors wrote that MSB reduced SCC but P>0.05.
Line 210: Figure 1, It is preferable to add standard deviation. The abbreviation should be added in the footnotes.
Conclusion
Even authors changed the conclusion, it is still very superficial. It does not shed light on the interest of the results obtained and the interest in using blocks with additives, and also it does not link them to the initial hypothesis reported in the introduction. This section needs an improvement.
Author Response
Response to Reviewer Comments
Reviewer# 3
Point 1: Abstract:
Line 18: The added introduction should present the importance to study milk production.
Response 1:
Your positive feedback was greatly appreciated. In that sentence, we did add milk production. Please refer to in line 19.
Point 2: Abstract:
Line 26: I think it is more convenient to change general by conventional.
Response 2:
As suggested, we changed ‘general’ to ‘conventional’, Please refer to in line 26.
Point 3: Abstract:
Line 40: authors should add a conclusion which presents the application of the results and recommendation.
Response 3:
We have included a suggestion for MSB supplementation. Please see line 40 for more information.
Point 4: Abstract:
Keywords: Selenium was not mentioned (authors focused on Bicarbonate de sodium more than Selenium)
Response 4:
In keywords, we replaced 'somatic cell count' with 'selenium' as suggested.
Point 5: Introduction:
The introduction should be improved. The authors did not present the interest in selenium and did not justify why they chose to add selenium.
Response 5:
As requested, we have included selenium in the introduction. For more information, please see in line 87-91.
Point 6: Material and methods
Line 100: in which country.
Response 6:
As a suggestion, we have added country, please refer to line 106.
Point 7: Line 166: “was analyzed” was reported twice in the sentence.
Response 7:
As suggested, we removed the phase ‘was analyzed’ from the sentence, please refer to line 171.
Point 8: Results
Line 180: delete the second “milk”
Response 8:
Line 180 as you suggested, we removed the ‘milk’ as suggested. Please see in line 184.
Point 9: Line 185: the authors wrote that MSB reduced SCC but P>0.05.
Response 9:
We wrote ‘When compared to the control, mineral salt block supplementation reduced SCC (P = 0.07) and the somatic cell score (SCS) (P < 0.05).’ Please see the text (line 188) for more information.
Point 10:
Line 210: Figure 1, It is preferable to add standard deviation. The abbreviation should be added in the footnotes.
Response 10:
We have added SE and put abbreviation of treatments as suggested. Please see figure 1 in the text for more information.
Point 11: Conclusion
Even authors changed the conclusion, it is still very superficial. It does not shed light on the interest of the results obtained and the interest in using blocks with additives, and also it does not link them to the initial hypothesis reported in the introduction. This section needs an improvement.
Response 11:
Thank you very much for your insightful remarks. Now we have modified as ‘According to the current findings, ruminal pH can be stabilized in the optimal range for rumen microbial activity and milk quality can be improved by decreasing somatic cells in milk by providing mineral salt blocks containing NaHCO3 or Se. How-ever, the current findings will need to be confirmed with a high-producing lactating dairy cow fed a high-concentrate diet, as well as an investigate into whether a mineral salt block could help maintain ruminal pH or increase milk output.’ We hope that the current revision of conclusion meets your requirements. Please in the manuscript. (line 320-325).
We thank you once again for your assistance.
Please see the attachment.

Round 3
Reviewer 3 Report
The manuscript presents the effect of the mineral salt blocks containing sodium bicarbonate or selenium on ruminal pH and volatile fatty acid, milk production, and composition, and blood metabolites.
I appreciate the huge effort and the changes made in the article. I feel it is more comprehensible and informative. All questions were answered and all recommended modifications were made.